IMC-YOLO: a detection model for assisted razor clam fishing in the mudflat environment

Xu Jianhao 1
Cao Lijie caolijie@dlou.edu.cn 1
Pan Lanlan 2
Li Xiankun 2
Zhang Lei 1
Gao Hongyong 1
Song Weibo swb@dlou.edu.cn 1
1 College of Information Engineering, Dalian Ocean University , Dalian , China
2 College of Mechanical and Power Engineering, Dalian Ocean University , Dalian , China
Alatas Bilal
Electronic publication date: 2025 Jan 10
Publication date: 2025
Volume: 11
Electronic Location ID: e2614
Received 2024 Jun 9; Accepted 2024 Nov 26
Copyright: ©2025 Xu et al.
Copyright year: 2025
Copyright holder: Xu et al.
License: This is an open access article distributed under the terms of the Creative Commons Attribution License, which permits unrestricted use, distribution, reproduction and adaptation in any medium and for any purpose provided that it is properly attributed. For attribution, the original author(s), title, publication source (PeerJ Computer Science) and either DOI or URL of the article must be cited.
License URL: https://creativecommons.org/licenses/by/4.0/

Keywords: Object detection, Intertidal mudflat culture, YOLOv8n, Feature fusion, Attention mechanism

Funding: Department of Education of Liaoning Province No. LJKM20221110 The Central Government Subsidy Project for Liaoning Fisheries (2024) This work was supported by the Department of Education of Liaoning Province (No. LJKM20221110) and The Central Government Subsidy Project for Liaoning Fisheries (2024). The funders had no role in study design, data collection and analysis, decision to publish, or preparation of the manuscript.

==============================
In intertidal mudflat culture (IMC), the fishing efficiency and the degree of damage to nature have always been a pair of irreconcilable contradictions. To improve the efficiency of razor clam fishing and at the same time reduce the damage to the natural environment, in this study, a razor clam burrows dataset is established, and an intelligent razor clam fishing method is proposed, which realizes the accurate identification and counting of razor clam burrows by introducing the object detection technology into the razor clam fishing activity. A detection model called intertidal mudflat culture-You Only Look Once (IMC-YOLO) is proposed in this study by making improvements upon You Only Look Once version 8 (YOLOv8). In this study, firstly, at the end of the backbone network, the Iterative Attention-based Intrascale Feature Interaction (IAIFI) module module was designed and adopted to improve the model’s focus on advanced features. Subsequently, to improve the model’s effectiveness in detecting difficult targets such as razor clam burrows with small sizes, the head network was refactored. Then, FasterNet Block is used to replace the Bottleneck, which achieves more effective feature extraction while balancing detection accuracy and model size. Finally, the Three Branch Convolution Attention Mechanism (TBCAM) is proposed, which enables the model to focus on the specific region of interest more accurately. After testing, IMC-YOLO achieved mAP50, mAP50:95, and F1best of 0.963, 0.636, and 0.918, respectively, representing improvements of 2.2%, 3.5%, and 2.4% over the baseline model. Comparison with other mainstream object detection models confirmed that IMC-YOLO strikes a good balance between accuracy and numbers of parameters.

Introduction

In the latter half of the last century, due to rapid advancements in fishing technology, the catch volume in fisheries significantly increased. Today, balancing the improvement of fishing efficiency with the reduction of environmental impact has become the development trend in the fishing industry (Valdemarsen, 2001). As a type of special wetland ecosystem (Van der Wegan, Roelvink & Jaffe, 2019), intertidal mudflat is one of the earliest marine territories successfully developed and utilized by human beings. Intertidal mudflat culture (IMC) is a form of aquaculture that utilizes intertidal mudflats for aquatic organism farming activities. As the key step of IMC, the fishing methods for aquatic organisms are crucial to the aquaculture process. As a marine mollusk of the class Bivalvia, razor clam Solen strictus is an important marine economic shellfish in East and Southeast Asia (Wang, Li & Guo, 2019). In 2021, the global total production of mollusks in aquaculture in marine regions reached 18.2 million tons (FAO, 2024). However, as a burrowing organism primarily inhabiting the intertidal zone, razor clams have traditionally been fished by fishers who rely on visual observation of the spatial distribution and quantity of razor clam burrows, followed by their collection using tools such as shovels. Specifically, fishers use a long, slender stick with a conical tip to probe into the holes, and they also use methods such as increasing the salinity of the environment to assist in the capture. This method of manual fishing suffers from inefficiency and is prone to various issues such as oversight. Compared to manual fishing methods, the use of large-scale excavation tools such as dredgers for extracting target shellfish from benthic organisms is more efficient. For example, razor clams can be fished using hydraulic dredging methods (Hauton et al., 2007). However, this mechanized method of fishing can have negative impacts on non-target species and seabed ecosystems and is a highly destructive fishing method (Murray et al., 2016). Therefore, there is an urgent need for an efficient and accurate sustainable razor clam fishing method for IMC. The prerequisite for realizing such a fishing method is to be able to accurately detect the spatial distribution and number of razor clam burrows before fishing.

Today, there is a widening gap between global fish supply and demand, and advanced aquaculture technology is the only way to address this challenge (Ashraf Rather et al., 2024). With the rapid development of artificial intelligence technology, the integration of computer vision and artificial intelligence (Lu, Guna & Dansereau, 2017) has been widely used in modern agriculture. Artificial intelligence is the ability of a system to solve advanced problems, and intelligent systems with this ability usually rely on machine learning (Janiesch, Zschech & Heinrich, 2021). And deep learning, as a branch of the machine learning field, is currently the most successful artificial intelligence technology (Strawn, 2022). In the field of modern aquaculture, artificial intelligence has shown enormous potential for application and provides many excellent solutions for agricultural practices such as sustainable aquaculture (Mandal & Ghosh, 2023). Meanwhile, smart aquaculture methods have already brought huge economic benefits to the aquaculture industry (Vo et al., 2021).

In recent years, as one of the fastest growing human activities (Tian & Li, 2023), aquaculture has been expanding, which in turn has led to an increasing contradiction between the complexity of the aquaculture environment and the refinement of the aquaculture process. In aquaculture, high-complexity scenarios are usually accompanied by a variety of potential factors that may affect detection accuracy, such as varying degrees of occlusion, large amounts of overlap, and small target sizes. These situations often lead to object detection algorithms in complex scenarios exhibiting lower accuracy and robustness than in other scenarios. For these issues, some scholars’ relevant studies can serve as references. For example, Wang et al. (2023). propose a reinforcement learning paradigm of configuring visual enhancement for object detection in underwater scenes. Siripattanadilok & Siriborvornratanakul (2023) achieved accurate detection of hidden crabs by using backbone network with different structures in Faster region-based convolutional ceural networks (R-CNN) (Siripattanadilok & Siriborvornratanakul, 2023). Zhang et al. (2024) achieved the task of accurately counting underwater fish under overlapping and occluded conditions by introducing soft non-maximum suppression algorithm (Soft-NMS) into the YOLOv5s-seg model based on BoTNet (BoTS-YOLOv5s-seg) model. He et al. (2024) combined the normalized Gaussian Wasserstein distance (NWD) with the intersection over union (IOU) inherent in the YOLOv5s according to a certain scale factor, and finally realized the effective detection and accurate measurement of small-sized aquatic products. To overcome visual degradation issues, Wang et al. (2024) introduces a thorough underwater image enhancement framework consisting of three stages: meta submergence, meta relief, and meta ebb. When performing object detection tasks, the image acquisition device needs to acquire sufficiently rich target information (Liu et al., 2023). Razor clam burrows are typically between a few millimeters and a few centimeters in diameter. This situation may result in individual razor clam burrow being too small to be accurately recognized by the detection algorithm when compared to the full image in a detection task targeting razor clam burrows. At the same time, the reflection of light by tides, the rise and fall of tides and the obstruction of targets by objects such as sand and gravel create a complex and variable mudflat environment. And this environment poses a serious challenge to the detection of razor clam burrows.

Currently, there is a gap in the application of object detection technology in razor clam fishing. To improve the efficiency of razor clam fishing, protect the ecological environment and promote the sustainable development of IMC, this study proposes an intelligent razor clam fishing method, which provides technical support for the realization of high-precision, high-efficiency and sustainable razor clam fishing. No one has yet established a valid dataset for razor clam fishing in modern aquaculture. To make up for the lack of effective datasets for razor clam fishing in the field of IMC, a razor clam burrows dataset with razor clam Solen strictus as the detection target was established in this study. Meanwhile, in order to adapt to the complex mudflat environment, an improved model called intertidal mudflat culture-You Only Look Once (IMC-YOLO) for IMC is proposed based on YOLOv8 Nano (YOLOv8n). Specific improvements include: combining the iterative Attention-based intrascale feature interaction (IAIFI) module and Conv-BN-SiLU (CBS) module to replace the Spatial Pyramid Pooling-Fast (SPPF) module in YOLOv8, which enhances the model’s ability to extract high-level features; reconfiguring head network, which enhances the model’s ability to detect small targets; realizing adequate maintenance of the model’s extracted features while reducing the model size through the introduction of the FasterNet block; proposing the hybrid Attention three branch convolution attention mechanism (TBCAM) to improve the model’s attention to key regions in the feature map.

Related work

Deep learning-based object detection algorithm

As one of the most fundamental and critical branches in computer vision, object detection techniques have been receiving attention from many researchers in the past decades (Zhao et al., 2019). Deep learning-based object detection algorithms usually have better performance and higher efficiency than traditional object detection methods. Currently, deep learning-based object detection methods are categorized into two types, namely, one-stage detection algorithms and two-stage detection algorithms (Deng et al., 2020). Among them, the early popular two-stage detection algorithm has a slow detection speed, which makes it difficult to adapt to the occasions requiring high detection speed (Redmon et al., 2015). Unlike two-stage detection algorithms, one-stage detection algorithms do away with the initial screening of candidate regions and instead utilize specific deep neural networks to directly predict the class and location of the target. Currently, one-stage detection algorithms have become the dominant algorithms in the field of object detection. One-stage detection algorithms such as SSD (Liu et al., 2016), CenterNet (Zhou, Wang & Krähenbühl, 2019) and YOLO series (Terven, Córdova-Esparza & Romero-González, 2023) have been widely used for real-time object detection and have become pioneering algorithms for other more advanced computer vision tasks such as object tracking, image characterization and event detection.

YOLOv8 model

In 2023, Ultralytics launched its latest open-source detection model, YOLOv8, to become the new state-of-the-art (SOTA) in object detection. The overall architecture of YOLOv8 model consists of three parts: the backbone network, the neck network and the head network. The design of the backbone network is similar to that of YOLOv5, using the DarkNet53 structure for high quality feature extraction. As shown in Fig. 1, the neck network adopts the feature fusion concept of Path Aggregation Network and Feature Pyramid Network (PAN-FPN), which is responsible for multi-scale fusion of a large amount of extracted feature information. The head network, unlike YOLOv5, adopts the anchor-free detection method, which directly predicts the center point information, making the processing more flexible and efficient. In addition, YOLOv8 abandons the coupled head and adopts the decoupled head, which separates the regression branch from the prediction branch and improves the independence of classification and regression. Also, depending on the scaling factor, YOLOv8 is available in five different sizes and complexity levels, namely YOLOv8n, YOLOv8s, YOLOv8m, YOLOv8l and YOLOv8x. Among them, YOLOv8n guarantees sufficient detection performance with small inference cost and relatively lightweight model size, making it more suitable for the outdoor environment where detection models need to be deployed to edge devices.

Figure 1 PAN-FPN structure.

Materials & Methods

Dataset construction

In this study, an IMC base in the area of Golden Pebble Beach in Dalian City, Liaoning Province, China, was used as the object of investigation. In order to avoid the problem of insufficient model generalization ability due to the single sample type and to adapt to the potential complexities in the mudflat environment, we chose to photograph the razor clam Solen strictus burrows on the mudflat from multiple distances and angles at different tidal heights. This strategy aims to ensure the comprehensiveness and diversity of data acquisition, to provide more representative and reliable data support for subsequent model training and analysis. Through the image collection process above, a total of 1,450 raw images were obtained. Images that were too blurred to annotate were manually removed, and due to their excessive size, the original images were all cropped into halves, resulting in a final set of 662 raw original images, each with a resolution of 1,080 × 960 pixels. Part of the original image is shown in Fig. 2.

Figure 2 Part of the original image.

Labelme, an image annotation tool (Torralba, Russell & Yuen, 2010) is used to annotate the original images acquired, and 70% of the annotated original dataset is used as the original training set. The image preprocessing process is crucial in the aquaculture field, and several researchers have conducted relevant studies. For example, Wang, Zhang & Ren (2024) develop a new reinforcement learning framework that autonomously selects and fine-tunes a sequence of image enhancement methods to enhance underwater images effectively. An adaptive attenuated channel compensation technique that leverages optimal channel precorrection and a salient absorption map-guided fusion approach to correct color deviations in the RGB color space was proposed (Wang, Sun & Ren, 2024).

To adapt to the different light intensity in the real mudflat environment, strengthen the anti-interference in the process of image acquisition, further improve the generalization ability of the model and prevent the overfitting phenomenon in the training process, data augmentation was performed by changing the image brightness, adding pretzel noise and flipping horizontally. Specifically, brightness can be adjusted in two ways: by increasing each color channel of the image by 40% to make it brighter, or by decreasing each color channel by 40% to make it darker. Additionally, the intensity of salt-and-pepper noise is set to 50%. To prevent the phenomenon of information leakage (Whalen et al., 2022), the data augmentation of the training set is performed in isolation from the data augmentation of the other datasets. The total number of samples in the dataset is finally extended to 3,310 and the ratio of training, validation and test sets is 7:1.5:1.5. The effect of different augmentation methods is shown in Fig. 3. As shown in Fig. 4, the small relative size of razor clam burrows, a detection target, in the dataset of this study poses a serious challenge to the accuracy of object detection.

Figure 3 Different data augmentation methods.

(A) Original image. (B) Higher brightness. (C) Lower brightness. (D) Pretzel noise. (E) Flipping horizontally.

Figure 4 Relative size distribution of detected targets in the dataset.

Introduction of IMC-YOLO

To improve the accuracy and robustness of object detection algorithms in the mudflat environment, this study proposes a detection model called IMC-YOLO based on YOLOv8. Its structure is shown in Fig. 5. Firstly, the additional CBS module and the IAIFI module proposed in this study are combined to replace the SPPF module at the end of the original backbone network, to strengthen the degree of attention to the high-level features, and to capture the location information by using absolute position coding. Secondly, the head network is refactored to enhance detection capabilities for small targets. Then, the FasterNet Block is introduced to replace Bottleneck in the CSP Bottleneck with 2 Convolutions (C2f) module, which achieves a balance between detection accuracy and model size. Finally, this study proposes a novel hybrid attention TBCAM, which fuses channel information and spatial information to strengthen the link between channel dimension and spatial dimension. The improved IMC-YOLO is used to improve the effectiveness and robustness of object detection algorithms for aquatic organism farming activities in the mudflat environment and to enhance the detection of small targets such as razor clam burrows.

Figure 5 IMC-YOLO model architecture.

IAIFI module

Intrascale feature interaction (AIFI) module is a key component of Real-Time DEtection TRansformer (RT-DETR) (Zhao et al., 2023). It utilizes a multi-head self-attention module in the transformer and integrates position information into the input features using absolute position coding based on sine and cosine positional encoding. This absolute position coding can be described in the Eq. (1): (1) PEx,2i= sinxT2idPEx,2i+1= cosxT2idPEy,2i= sinyT2idPEy,2i+1= cosyT2id

where x and y denote the position indexes of the element in width and height respectively, d denotes the dimension of the embedding vector, i denotes the index of the dimension of the embedding vector, and T is the set temperature parameter. By specifying the value of T, the period of the trigonometric function in the above equation can be adjusted to regulate the coding granularity more flexibly. From the above equation, the embedding vector of position (x, y) can be obtained, and by concatenating the embedding vectors of x and y, the rich position information of different frequencies can be captured to form the final position encoding.

The absolute position encoding method in the AIFI module avoids the dependence on the convolutional kernel and achieves a more flexible approach to feature processing than methods that rely on convolutional operations to obtain contained position information. The internal processing flow of the AIFI module is summarized in the Eq. (2): (2) Q=K=V=FlattenFin

(3) Fout=ReshapeAttnQ,K,V

(4) AttnQ,K,V=SoftmaxQKTdk×V

where the input tensor Fin is subjected to the operation of reshaping, resulting in the generation of query vector Q, key vector K, and value vector V for multi-head attention computation. Subsequently, in the Eq. (3), the model captures positional dependencies and higher-level semantic information through the computation of multi-head attention. The self-attention mechanism is illustrated as shown in the Eq. (4): Q represents the query matrix, K denotes the key matrix, V stands for the value matrix, and dk refers to the dimension of Q and K. The multi-head attention module is composed of multiple self-attention modules, which is given by the Eq. (5) and the Eq. (6): (5) mAttnQ,K,V=Concathead1,…,headnWO

(6) head=AttnQWiQ,KWiK,VWiV

where the WO, WiQ , WiK and WiV is the projections parameter matrices. Following this, an inverse operation is applied to reshape the tensor back to its original shape, yielding the final output tensor Fout.

Skip connection is a widely used operation in modern convolutional neural networks and is usually limited to two of the feature fusion methods: concatenation and addition. However, these two feature fusion methods ignore variations in the features themselves, giving the same weight to two tensors with different content. Therefore, this study introduces the iterative Attentional Feature Fusion (IAFF) (Dai et al., 2021). The feature fusion method of IAFF is shown in Fig. 6. The Multi-Scale Channel Attention Module (MS-CAM) within IAFF enables channel attention at multiple scales by varying the pooling scale. Specifically, MS-CAM employs global average pooling in the global attention branch, and the two branches undergo two point-by-point convolutions, as well as intermediate activation and batch normalization layers, respectively, which are element-wise addition to obtain the fusion features through the broadcasting mechanism, and the Sigmoid activation function to obtain the channel attentions ranging from 0 to 1. For the original input tensor X and the output tensor Y after a series of transformations, the channel attention α output from the MS-CAM is used as the weight of X, and (1- α) is used as the weight of Y. X and Y are weighted and added element-wise. IAFF sequentially performs two rounds of the above operations, and with the high-quality fusion features generated in the first round as the initial features, the fusion weights in the second round are sufficiently improved, which in turn realizes the full context-awareness and the final output tensor Z is obtained.

Figure 6 Illustration of the IAFF.

To extract deep features rich in more advanced semantic information and at the same time to enhance the model’s representational capability, this study adopts the IAFF-based skip connection operation to transform the AIFI module. The modified module called IAIFI is combined with the CBS module to replace the SPPF module in YOLOv8 model. As shown in Fig. 7, Firstly, with the help of the additional CBS module, the model can further extract the features in the original backbone network. Subsequently, the IAIFI module relies on the multi-head self-attention mechanism of the high-level features to further enhance the interactions among the high-level features, to obtain the correlations among the conceptual entities in the image. This design approach, which focuses on establishing relationships between different regions within low-resolution feature maps, allows the model to learn abstract correlations of a single razor clam burrow relative to the entire image. This helps in distinguishing the regions containing razor clam burrows from similar but non-target regions, thereby reducing the false detection rate of razor clam burrows in practical applications. Finally, it relies on IAFF to perform skip connection operations on the input tensor and output tensor. The significance of this design strategy is to achieve a more rational fusion of semantically and scale-inconsistent features during skip connection operations.

Figure 7 Illustration of the IAIFI.

Refactoring head network

The head network of the YOLOv8 model consists of three different scales of feature maps suitable for recognizing large, medium and small targets. However, for the object detection task, the pixels occupied by a single razor clam burrow in the input image in the context of a mudflat environment are extremely limited, resulting in the three basic feature map scales being difficult to meet the recognition and regression requirements of the task. Therefore, IMC-YOLO introduces a larger feature map scale by adding an Upsampling module and a Concat module, and realizes accurate recognition of small targets such as razor clam burrows by deep fusion of features with other scales.

In the original YOLOv8 model, the fusion of features at different scales relied on the PAN-FPN structure of the neck network. However, the head network of YOLOv8 does not fully utilize features at different scales as inputs to the head network. Therefore, in order to obtain feature information at different levels and granularities, as well as to mitigate the inconsistency between the four different feature map scales, the adaptively spatial feature fusion (ASFF) (Liu, Huang & Wang, 2019) method was used in this study to improve the head network in IMC-YOLO. The value of this strategy is that, according to different spatial locations and scales, the ASFF method can dynamically adjust the feature weights at each scale, thus achieving a balance between feature information of different scales and importance. Specifically, during razor clam burrows detection, variations in shooting distance and angle inevitably cause shifts in features of different scales, which can impact the importance of these features. To mitigate this issue, we choose to enable the network to implicitly learn the importance of features at different scales, achieving a certain level of alignment for these features. The ASFF method relies on multiple convolutional layers and the final SoftMax function to dynamically learn the feature weights at different scales and obtain the final feature map through weighted fusion. The specific feature fusion formula is as the Eq. (7) and the Eq. (8): (7) yijl=αijlxij1→l+βijlxij2→l+γijlxij3→l+δijlxij4→l

(8) αijl=eλαijleλαijl+eλβijl+eλγijl+eλδijl

where xijf→p denotes the feature mapping of the f-scale feature at position (i, j) with respect to the p-scale feature. αijl, βijl, γijl, and δijl are the weights of the feature mappings of the different scales at position (i, j) learned by the SoftMax function in the Eq. (8), which take the value range of (0, 1), and αijl + βijl + γijl + δijl = 1. yijl is the final feature obtained by weighted fusion at position (i, j).

FasterNet block

In aquatic organism farming activities, models usually need to be deployed to mobile devices with limited resources, and in this application scenario, the size of the model will be very limited. Therefore, in order to improve the applicability of the model in the above cases, this study introduces the FasterNet Block (Chen et al., 2023), to replace the Bottleneck in the C2f module, which improves the feature extraction capability of the model while realizing a certain reduction in the size of the improved model.

FasterNet Block employs a new lightweight convolution called PConv, as shown in Fig. 8, which achieves more efficient feature extraction while reducing the model size. In a conventional convolution operation, all input channels are involved in the computation, which leads to a large number of redundant computations. To solve this problem, PConv performs the traditional convolution operation only on some of the consecutive input channels, leaving the rest of the channels unchanged. The reason this approach is similar to, but different from, the direct deletion of some of the channels is that it ensures that the feature information can flow through all the channels in subsequent operations. By reducing the involvement of redundant channels in the model, we have minimized the risk of overfitting that may arise from model complexity and alleviated the demand on storage capacity during actual fishing operations. As shown in Fig. 9, compared to Bottleneck, the FasterNet Block adds the PConv operation, a convolutional kernel size of 3 ×3, an MLP structure implemented by two point-by-point convolutional layers with intermediate activation and batch normalization layers, and a skip connection operation to improve the network performance for all C2f modules of different depths.

Figure 8 Illustration of the PConv.

Figure 9 Comparison of Bottleneck and FasterNet Block.

(A) Bottleneck. (B) FasterNet Block.

TBCAM

In the field of computer vision, the attention mechanism has been widely used in various tasks such as semantic segmentation, instance segmentation and object detection. Attention mechanisms are flexible in feature processing and enable the model to better focus on key regions in the image, thus improving the accuracy of image processing tasks. Currently, hybrid attention mechanisms such as CBAM (Woo et al., 2018), GAM (Liu, Shao & Hoffmann, 2021) and TripletAttention (Misra et al., 2020) emphasize simultaneous attention to channels as well as spatial regions. This hybrid attention mechanism utilizing channel and spatial information enables the model to better understand the semantic structure and spatial layout of images. Therefore, to improve the model’s ability to process image features in the mudflat environment, this study proposes a hybrid attention TBCAM whose complete structure is shown in Fig. 10.

Figure 10 Structure of the TBCAM.

For the channel level, in order to enhance the feature expressiveness of each channel, TBCAM performs alternating average pooling and maximum pooling on the height and width dimensions of the feature map, respectively. Compared with single average pooling or maximum pooling, the alternate pooling strategy yields more effective and critical information in the width and height dimensions. Subsequently, the obtained features are fed separately into a shared MLP to learn the complex relationships between the channels. By adding the two copies of the output features after the above process and using a Sigmoid activation function to learn the nonlinear feature representation capabilities, a representation of the weights of each channel is obtained. Eventually, these channel weights are multiplied element-wise with the original input features, which is the complete process of TBCAM at the channel level.

As for spatial dimensions, traditional spatial attention tends to focus only on the intrinsic connection of spatial dimensions, ignoring the potential relationship between channel dimensions and spatial dimensions. Therefore, to utilize this potential relationship, TBCAM borrows the three-branch structure in TripletAttention, which establishes inter-dimensional dependencies by performing a dimension-level rotation operation on each branch, thus capturing cross-dimensional interactions of input features. Specifically, three branches of TBCAM are used to construct the link between the height and width dimensions, the link between the channel dimension and the height dimension, and the link between the channel dimension and the width dimension. For the channel-level output, the tensor x ∈ RC×H×W will be passed to each of the three proposed branches. In the first branch, the maximum and average values of the tensor along the channel dimension are first computed and concatenation are performed. Subsequently, the weight of each feature point in the feature map is obtained by a convolution operation with a convolution kernel size of 7 × 7 without changing the dimension of the feature map and using batch normalization and Sigmoid activation function. Finally, the final output of the first branch is obtained by multiplying element-wise these weights with the original feature map. For the remaining two branches used to construct the dependencies between the spatial and channel dimensions, it is necessary to permute the dimensions to obtain the tensor y1 ∈ RW×H×C and the tensor y2 ∈ RH×C×W, respectively, and then take the same operation as in the first branch and permute the dimensions with their original states. The final output tensor is obtained by element-wise addition of the output tensors of the three branches described above, followed by a simple averaging calculation.

Eventually, this study embedded the TBCAM in the FasterNet Block before making a skip connection operation to obtain the Three Branch Attention Faster Net (TBAFN) module as shown in Fig. 11, which led to the improved module called C2f_TBAFN. The TBCAM allows the model to focus more on key areas related to razor clam burrows in the image, effectively reducing the impact of background noise and interference. This not only enhances the model’s ability to identify the target but also improves its performance when handling images with varying scales and complex backgrounds.

Figure 11 Comparison of FasterNet Block and TBAFN.

(A) FasterNet Block. (B) TBAFN.

Experimental platform and experimental hyperparameters

The experiments in this article were conducted under the Windows 10 operating system environment (Microsoft, Bellingham, WA, USA). The computers used for the experiments were configured with a 12th generation Intel (R) Core (TM) i7-12700KF CPU @ 3.60 GHz, 32GB of RAM, and a NVIDIA GeForce RTX 4070Ti GPU with 12GB of video memory. For the compilation environment, the experiments were conducted using Python 3.9 + PyTorch 2.2.1 + Compute Unified Device Architecture (CUDA) 12.1.

To obtain the best model performance, the relevant hyperparameters for model training were appropriately adjusted in this study, and the detailed hyperparameter settings are shown in Table 1.

Table 1 Experimental hyperparameters.

Hyperparameters	Value	
Epoch	100	
Batch size	16	
Input image size	(640,640)	
Optimizer	Adam	
Initial learning rate	0.001	
Scale factor for final learning rate	0.01	

Evaluation indicators

There are some generalized metrics for evaluating object detection models in object detection tasks. In the razor clam burrows detection task, to obtain the model’s detection accuracy for razor clam burrows under different conditions, and assess the model’s storage resource requirements for practical application, the mean average precision (mAP), the best F1 score (F1best), and the number of parameters (Params) are used as the evaluation metrics of the model. Among them, two metrics, mAP and F1best, are used to test the detection effect of the model. The metrics are defined as the Eqs. (9)–(13): (9) P=TPTP+FP

(10) R=TPTP+FN

(11) AP= ∫01PRdR

(12) mAP=1QR ∑q=QRAPq

(13) F1=2TP2TP+FP+FN

where TP is the number of positive samples in the category predicted as positive by the model, FP is the number of negative samples in the category predicted as positive by the model, FN is the number of positive samples in the category predicted as negative by the model, and TN is the number of negative samples in the category predicted as negative by the model. Indicator P denotes the proportion of samples predicted correctly out of those predicted positive, which is referred to as precision rate. Indicator R indicates the proportion of samples predicted correctly among all positive samples, known as recall rate. Indicator AP denotes average precision, i.e., the area under the precision–recall (P-R) curve for a single category, and QR denotes the total number of categories. The metric mAP evaluates the performance of the model on multiple categories by calculating the average of the mean precision rates of the different categories. F1best denotes the harmonic mean of P and R at the optimal threshold, which is used to gain the recall and precision rates of the model.

In razor clam fishing, mAP can assess the model’s basic accuracy in detecting razor clam burrows. Improving this metric indicates that the model can accurately identify clam holes at lower detection standards. F1best, on the other hand, considers both precision and recall, evaluating the overall detection quality of the model. It helps balance the situations of false positives (incorrectly detecting something as a clam hole) and false negatives (failing to detect an actual clam hole), thereby enhancing the accuracy and reliability of the detection.

Results

Experimental results

In general, the edge devices used to deploy the models have low storage capacity and computational power. Therefore, the smallest of the different versions of YOLOv8, YOLOv8n, was chosen as the baseline model in this study. The complete process of detecting razor clam burrows is shown in Fig. 12. First, after acquiring the original images, the razor clam burrows dataset constructed in this study was processed using various data augmentation. Then, the IMC-YOLO proposed is constructed and set appropriate experimental hyperparameters for training. Eventually, IMC-YOLO, which completed the training, can realize the recognition and number counting of razor clam burrows in mudflat environments. Figure 13, demonstrates the training of the baseline model and IMC-YOLO, as can be seen from the figure, the mAP50 as well as the mAP50:95 of the validation set show a smooth incremental trend as the epoch increases. Starting from the 40th epoch, the detection effect of IMC-YOLO is completely higher than that of the baseline model, which indicates that the trained IMC-YOLO can learn more effective and richer feature information. From the 60th epoch onwards, the mAP50 and mAP50:95 of IMC-YOLO still shows a small increase compared to the converged baseline model, which indicates that for the high-quality features extracted by IMC-YOLO, the model has more room for learning, and thus can obtain more accurate detection results.

Figure 12 Process of detecting razor clam burrows.

Figure 13 Comparison of IMC-YOLO and YOLOv8n training situations.

The test set images are input into the baseline model and IMC-YOLO, respectively, and the prediction results are obtained after inference. The results containing quantitative information are finally obtained according to the predicted number of similar anchor frames in the prediction results. A few examples of the prediction results are shown in Fig. 14. The baseline model misses four targets in the first case and one target in the second case, which will reduce the efficiency of razor clam fishing to some extent. Compared with the baseline model, the IMC-YOLO misses only one target in the first case and detects all targets in the second example, which satisfies the prerequisite for efficient razor clam fishing.

Figure 14 Predicted results.

(A) Ground truth. (B) YOLOv8n. (C) IMC-YOLO.

Figure 15 Comparison of the area under the P-R curve.

Meanwhile, there is a certain degree of dynamic blurring in the first case, while IMC-YOLO can realize accurate identification of razor clam burrows in this case, which provides a safeguard against abnormalities such as shaking of the catching equipment that may occur during razor clam fishing. At the same time, IMC-YOLO can accurately detect targets that are difficult to detect due to reflected light from the water surface, thus proving that IMC-YOLO is well adapted to the mudflat environment. In summary, the IMC-YOLO shows obvious advantages in the application of razor clam fishing.

Precision and recall, as a set of mutually constraining evaluation metrics, can only improve the overall performance of the model more effectively and comprehensively if the simultaneous improvement of the two is realized. Therefore, the P-R curves of IMC-YOLO and the baseline model are plotted in this study. As shown in Fig. 15, relative to the baseline model, the area under the P-R curve of IMC-YOLO is larger and completely covers the baseline model, which suggests that IMC-YOLO will have a more balanced and better detection in the same situation. Based on the above conclusions, the detection performance of the IMC-YOLO proposed in this study was effectively validated.

Ablation experiments

To validate the effectiveness and feasibility of the improvement part proposed in the task of object detection in the mudflat environment, the ablation experiments shown in Table 2 were designed and conducted in this study. A total of five sets of experimental models were obtained and validated by gradually adding the improvement part proposed to the baseline model, where YOLOv8n is the baseline model, serial Module 1–3 are the models improved by adding this work respectively and IMC-YOLO is the final improved model obtained. At the same time, we also removed the IAIFI module and the refactoring of head network separately from IMC-YOLO, resulting in two additional experimental models, namely Module 4 and Module 5.

Table 2 Results of ablation experiments.

Model	IAIFI	Refactoring head network	FasterNet block	TBCAM	mAP50 (%)	mAP50:95 (%)	Params (10 6 )	F1best	
YOLOv8n	–	–	–	–	0.941	0.601	3.01	0.894	
Model1	✓	–	–	–	0.946	0.610	2.95	0.906	
Model2	✓	✓	–	–	0.949	0.614	4.26	0.914	
Model3	✓	✓	✓	–	0.956	0.630	3.54	0.916	
Model4	–	✓	✓	✓	0.952	0.625	3.60	0.915	
Model5	✓	–	✓	✓	0.956	0.622	2.26	0.912	
IMC-YOLO	✓	✓	✓	✓	0.963	0.636	3.50	0.918	

Based on the experimental results, each kind of addition has a certain enhancement effect on each metric. Firstly, through the combination of CBS module and IAIFI module, the mAP50, mAP50:95 and F1best both increase to a certain extent. Subsequently, refactoring head network also has some improvement in each metric, but with the number of participants the number of senators has risen. To address this issue, the introduction of FasterNet Block reduces the number of parameters in the model, which helps IMC-YOLO to balance the accuracy and model size. Finally, the TBCAM is added to the model, which realize the overall improvement in detection results. Additionally, the experiments that removed the improvement components further demonstrated that each modification is crucial for the detection performance of the complete model. The above improvements achieved mAP50, mAP50:95, and F1best of 0.963, 0.636, and 0.918, respectively, representing improvements of 2.2%, 3.5%, and 2.4% over the baseline model.

To further investigate the impact of the proposed improvements on the detection performance of bamboo clam burrows at different sizes and distances, we conducted experiments on targets of various sizes and categorized the targets based on the pixels occupied by the detection box. Specifically, we defined targets with an area smaller than 32 pixels as small targets, those between 32 and 96 pixels as medium targets, and those larger than 96 pixels as large targets. By removing each of the four proposed improvements, we obtained the experimental results shown in Table 3. It can be observed that for small targets, IMC-YOLO achieved a 5.2% improvement in mAP50:95 compared to the baseline model. The removal of the reconstruction detection head had the largest negative impact on small target detection, reducing the mAP50:95 by 3.2%. This indicates that the multi-scale feature fusion in the small-size detection head significantly improves small target detection performance. Additionally, the introduction of the FasterNet Block also had a significant impact on small target detection, as PConv reduces redundant computation by limiting the number of channels involved, forcing the model to learn more effective features for small targets while ignoring more noise features, thus effectively constraining the model’s implicit learning of small target features. Compared to the baseline model, IMC-YOLO achieved a 2.8% and 2.4% improvement in mAP50:95 for medium and large targets, respectively, with these improvements being lower than for small targets, demonstrating that IMC-YOLO is more suitable for detecting small-sized and distant razor clam burrows.

Table 3 Experiments with targets of different sizes.

Model	IAIFI	Refactoring head network	FasterNet block	TBCAM	Small	Medium	Large	
YOLOv8n	–	–	–	–	0.494	0.645	0.744	
Model A	–	✓	✓	✓	0.533	0.664	0.753	
Model B	✓	–	✓	✓	0.514	0.67	0.763	
Model C	✓	✓	–	✓	0.52	0.663	0.755	
Model D	✓	✓	✓	–	0.538	0.67	0.743	
IMC-YOLO	✓	✓	✓	✓	0.546	0.673	0.768	

Comparative experimental analysis

To further validate the detection effect of IMC-YOLO in the mudflat environment, IMC-YOLO is selected for comparison with some mainstream object detection models in this paper. The specific experimental results are shown in Table 4. From the experimental results, it can be seen that in terms of the number of parameters, although the model size of YOLOv5n is the smallest, its coupled head does not realize the separation of the classification task and the regression task, which will largely affect the classification effect in the detection task, leading to difficulties in adapting to the subsequent study of carrying out the detection task for the multi-category targets containing razor clams, razor clam burrows, and other intertidal aquaculture organisms. Meanwhile, although IMC-YOLO has a slightly larger number of parameters compared with the baseline model, it has the highest mAP50 and mAP50:95, which means that IMC-YOLO has the best detection performance among the mainstream detection models experimented. In other words, IMC-YOLO achieves an effective improvement in the model performance at the cost of aslight increase in the number of parameters. These findings suggest that IMC-YOLO strikes an optimal balance between detection effectiveness and model size.

Table 4 Comparison of experimental results.

Model	mAP50 (%)	mAP50:95 (%)	Params (10 6 )	
YOLOv8n	0.941	0.601	3.01	
Faster R-CNN (Resnet18)	0.894	0.577	28.28	
CenterNet (Resnet18)	0.928	0.565	19.09	
YOLOv5n	0.937	0.606	2.50	
YOLOv3-tiny	0.911	0.574	12.13	
YOLOv7-tiny	0.921	0.59	6.00	
YOLOv10n	0.937	0.635	2.69	
IMC-YOLO	0.963	0.636	3.55	

Performance in special scenarios detection

The study also conducted experiments on the simulated data in the test set, targeting scenarios with low and high light intensity under realistic conditions, as well as scenarios where abnormal noise occurs during data transmission. The specific experimental results are shown in Table 5. Compared to the baseline model, IMC-YOLO demonstrates a uniform improvement in detection performance across three special scenarios, with mAP50:95 increasing by 3.9%, 3.4%, and 2.0%, respectively. These improvements indicate that IMC-YOLO still performs well in these special scenarios. Notably, the most significant enhancement in detection capability is observed in high and low illumination scenarios, which suggests that IMC-YOLO can maintain high detection accuracy even in scenes with significant lighting variations or suboptimal lighting conditions. This performance proves that IMC-YOLO can still effectively detect razor clam burrows in diverse and complex scenarios, thereby effectively assisting fishing activities.

Table 5 Experimental results under different abnormal scenarios.

Model	Low light	High light	Abnormal noise	
YOLOv8n	0.597	0.587	0.599	
IMC-YOLO	0.638	0.621	0.619	

The analysis of embedding location for TBCAM

To investigate the effect of different locations of TBCAM in IMC-YOLO on the model performance, three sets of comparison experiments based on different locations were designed. The experimental results are shown in Table 6, which shows that, through embedding the TBCAM into the FasterNet Block at the location before the skip connection, the model can obtain the best mAP50, mAP50:95, and F1best. The reason is that, as a key part of the feature extraction model, C2f_TBAFN achieves better maintenance of the extracted features while controlling the model size by stacking multiple FasterNet Blocks. Therefore, embedding TBCAM within FasterNet Blocks allows the model to maximally focus on regions of interest and achieve multi-level, fine-grained feature optimization. In contrast, adding TBCAM at the end of the backbone network, i.e., after the IAIFI module, does not allow for more detailed optimization of features at different depths. For embedding the TBCAM after each C2f of the neck network, the module has completed feature extraction before attention-based optimization is performed, and this lag can result in the TBCAM not being able to take full advantage of its role. Therefore, this study chooses to embed the attention mechanism of TBCAM into FasterNet Block before performing skip connections, so that the model can pay more attention to the key areas in the feature map and provide an effective guide for accurately identifying razor clam burrows.

Table 6 Impact of TBCAM location on model performance.

Location of TBCAM	mAP50 (%)	mAP50:95 (%)	F1best	
Inside FasterNet block	0.963	0.636	0.918	
After IAIFI module	0.960	0.631	0.915	
After each C2f in part neck network	0.960	0.635	0.916	

Experiments on detection effects on public datasets

To further validate the detection performance of the IMC-YOLO proposed in this study in the field of IMC, comparative experiments were designed using publicly available datasets for both the baseline model and IMC-YOLO. Shellfish are the main organisms in IMC. Therefore, this study adopts the publicly available dataset Shells (Roboflow, Inc, 2024) provided by Robflow, whose annotation categories are shellfish with different colors. Given that this study aims to construct a detection method applicable to IMC, and pure color has no practical significance forIMC, this study considers different colored shellfish in the dataset as the same category and excludes the segmented annotated data and unlabeled data from the dataset. Finally, a total of 1,490 images and the corresponding annotations are obtained, of which the numbers of the training set, the validation set, and the test set are 1300, 127 and 63, respectively, and some of the images are shown in Fig. 16.

The experimental results are shown in Fig. 17, and the mAP50, mAP50:95 and F1best of IMC-YOLO are improved by 6.6%, 5.1% and 4.4%, respectively, compared with the baseline model. This indicates that IMC-YOLO can identify the target more accurately in the mudflat environment and then provide a good guide for relevant detection tasks in IMC.

Figure 16 Partial shellfish image.

Figure 17 Experimental results on public datasets.

Discussion

In the porcess of IMC, it is often difficult to accurately identify targets due to the complex and variable topography of the mudflat environment, different lighting conditions and high and low tides. In this study, we introduce the object detection method to the razor clam fishing process and provide, which is suitable for detecting razor clam burrows, thus extending the applicability of the object detection technique to the mudflat environment.

In the field of IMC, the introduction of object detection technology not only aids in accurately identifying razor clam burrows but also significantly improves overall aquaculture management. By combining high-resolution imaging with deep learning algorithms, it can address the complex terrain and lighting conditions of intertidal zones, thus enhancing the accuracy and reliability of detection. Furthermore, these technologies can support real-time monitoring, providing immediate feedback that helps quickly adjust aquaculture strategies and optimize operations. Intelligent object detection can also be applied to monitor other economically valuable aquatic species, such as sea snails and clams, thereby broadening its application scope.

Conclusions

In order to improve the efficiency of razor clam fishing and guarantee the sustainable development of IMC, this study proposes a model called IMC-YOLO for razor clam burrows detection in the mudflat environment, which enhances the detection capability of small targets by refactoring the head network, and at the same time combines with the IAIFI module and C2f_TBAFN module proposed in this study, which achieves high-quality feature extraction and multiscale feature fusion. In addition, a hybrid attention TBCAM is proposed to improve the model’s attention to key regions in the feature map.

In this study, by detecting razor clam burrows in real time and counting the number of burrows in the detection area, we can more accurately understand the number and relative density of razor clams in subsequent razor clam fisheries, which can help to formulate the management strategy of IMC. Ablation and comparative experiments are designed to demonstrate the applicability of IMC-YOLO in the mudflat environment. By analyzing the impact of different depth features on model performance, TBCAM is able to maximize fine-grained feature optimization, effectively distinguishing between background and targets.

In the future, integrating sensor data with machine vision technology will contribute to developing more intelligent aquaculture systems capable of automatically identifying, classifying, and harvesting target organisms. The use of robotic arms will enable efficient operations with minimal human intervention, reducing labor intensity and improving operational precision. Overall, these technological advancements will drive the aquaculture industry towards more efficient and environmentally friendly practices, promoting the realization of sustainable farming models.

The authors express their gratitude to the editor and reviewers for their valuable comments and suggestions, which significantly contributed to the improvement of this manuscript.

Additional Information and Declarations

Competing Interests

Author Contributions

Data Availability

The authors declare there are no competing interests.

Jianhao Xu conceived and designed the experiments, performed the experiments, analyzed the data, performed the computation work, prepared figures and/or tables, authored or reviewed drafts of the article, and approved the final draft.

Lijie Cao conceived and designed the experiments, authored or reviewed drafts of the article, and approved the final draft.

Lanlan Pan analyzed the data, authored or reviewed drafts of the article, and approved the final draft.

Xiankun Li analyzed the data, prepared figures and/or tables, and approved the final draft.

Lei Zhang performed the experiments, analyzed the data, authored or reviewed drafts of the article, and approved the final draft.

Hongyong Gao conceived and designed the experiments, performed the experiments, performed the computation work, prepared figures and/or tables, and approved the final draft.

Weibo Song conceived and designed the experiments, analyzed the data, authored or reviewed drafts of the article, and approved the final draft.

The following information was supplied regarding data availability:

The code is available at Zenodo: Glenn Jocher, Ayush Chaurasia, Laughing, Muhammad Rizwan Munawar, Abirami Vina, Burhan, Lakshantha Dissanayake, Sergiu Waxmann, Colin Wong, Ivan Shcheklein, Kayzwer, Onuralp SEZER, WangQvQ, Adrian Boguszewski, triple-mu, Yonghye Kwon, Kalen Michael, DennisJ, Andy, …Sulthan Suresh Fazeela. (2024). MrGuanYu/IMC-YOLO: imcv2 (Version v2). Zenodo. https://doi.org/10.5281/zenodo.11482501.

The dataset is available at Zenodo: Xu, J. (2024). razor_clam_burrows [Data set]. Zenodo. https://doi.org/10.5281/zenodo.11480934.

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
