# Peer review of "IMC-YOLO: a detection model for assisted razor clam fishing in the mudflat environment"

_PeerJ Computer Science, doi:10.7717/peerj-cs.2614_

## Round 0.1 · original submission · Major Revisions

Dear authors,

Thank you for submitting your article. Based on reviews' comments, your article has not yet been recommended for publication in its current form. However, we encourage you to address the concerns and criticisms of the reviewers and to resubmit your article once you have updated it accordingly. Before submitting the paper, following should also be addressed:

1. Equations should be used with correct equation number. Please do not use “as follows”, “given as”, etc. Explanation of the equations should also be checked. All variables should be written in italic as in the equations. Their definitions and boundaries should be defined. Necessary references should be provided.
2. Pay special attention to the usage of abbreviations. Spell out the full term at its first mention, indicate its abbreviation in parenthesis and use the abbreviation from then on.
3. Please indicate practical advantages, discuss research limitations, and include future research directions.
4. Reviewer 2 has advised you to provide specific references. You are welcome to add them if you think they are useful and relevant. However, you are under no obligation to include them, and if you do not, it will not affect my decision.

Best wishes,

Reviewer 1 ·

Basic reporting

This paper introduces an intelligent method for razor clam fishing, aiming to improve efficiency and reduce environmental damage by employing a novel detection model, IMC-YOLO, based on YOLOv8. The study claims improvements in detection accuracy and model performance. While the research addresses a pertinent issue, several critical aspects require substantial improvement to enhance the overall quality and reliability of the findings.
1) There is a lack of comprehensive literature review. Discuss existing methods for razor clam fishing and previous attempts to integrate technology into this activity. This will help position your study within the broader research context.

Experimental design

2) The description of the razor clam burrows dataset is inadequate. Provide detailed information on how the dataset was collected, including the number of samples, geographic locations, and any preprocessing steps undertaken.

3)The improvements made to YOLOv8 (IAIFI module, refactored head network, FasterNet Block, and TBCAM) need more detailed explanation. Justify the choice of these modifications and describe how they specifically address the challenges in detecting small targets like razor clam burrows.

4) There is a lack of clarity on the training process, including hyperparameters, training duration, and computational resources used. This information is crucial for reproducibility.

Validity of the findings

5) The paper presents mAP50, mAP50:95, and F1best as performance metrics. Explain why these metrics were chosen and how they reflect the model's performance in practical scenarios.

6) The comparison with the baseline model is mentioned but lacks depth. Provide detailed quantitative and qualitative comparisons, highlighting specific scenarios where IMC-YOLO outperforms the baseline.

7) The comparison with other mainstream object detection models is superficial. Include a comprehensive analysis with statistical significance testing to support claims of superiority.

8) Discuss the feasibility of implementing IMC-YOLO in real-world razor clam fishing activities. Address potential challenges and limitations in terms of cost, scalability, and maintenance.

Reviewer 2 ·

Basic reporting

1. The current version contains many grammar errors. The authors are suggested to improve the presentation quality carefully.
2. The manuscript primarily focuses on object detection and requires the inclusion of the latest literature review on object detection, e.g., 'A Reinforcement Learning Paradigm of Configuring Visual Enhancement for Object Detection in Underwater Scenes. IEEE Journal of Oceanic Engineering'. Additionally, image pre-processing is a critical component of the detection process proposed in the manuscript, necessitating the inclusion of the latest literature review on image enhancement, e.g., 'Self-organized Underwater Image Enhancement. ISPRS Journal of Photogrammetry and Remote Sensing', 'Metalantis: A Comprehensive Underwater Image Enhancement Framework. IEEE Transactions on Geoscience and Remote Sensing', and 'Underwater Color Disparities: Cues for Enhancing Underwater Images toward Natural Color Consistencies. IEEE Transactions on Circuits and Systems for Video Technology'.

Experimental design

3. The proposed method in the manuscript involves an improvement to YOLOv8. Given that YOLOv10 has been released, the experimental section should include a comparison with the latest detector.
4. The comparison between computational efficiency (e.g., time) needs to be conducted to show the property of the proposed method better.

Validity of the findings

5. In the Conclusion section, authors can add outlooks for future work.

Reviewer 3 ·

Basic reporting

Manuscript is well written using scientific English language. Objectives and methodology are clearly defined. Background and literature survey is adequate. References used are from relevant indexed journals and conferences. Script is supported by neatly drawn figures and result tables.

Experimental design

It indicates original research. One major contribution is creation and annotation of dataset of 662 images of razor clam Solen strictus burrows on the mudflat from multiple distances and angles at different tidal heights.
To improve the efficiency of razor clam fishing and at the same time reduce the damage to the natural environment, an intelligent razor clam fishing method is proposed. Methodology is described with support of detailed description of various deep learning architectures, modification at different layers. It is also supported by relevant mathematical treatment. A modified version of YOLOv8 called IMC-YOLO is proposed. Hybrid attention mechanism is used to focus on key regions in the image.
Various deep learning models are experimented for the detection of razor clam burrows and rigorous investigation is performed.

Validity of the findings

Authors proposed a system to detect razor clam burrows in real time and counting the number of burrows in the detection area. This will help in more accurately understanding the number and relative density of razor clams in subsequent razor clam fisheries, which can help to formulate the management strategy of intertidal mudflat culture.
All underlying data have been provided; they are robust, statistically sound, & controlled.
Results, Discussions and conclusions are well stated.

Additional comments

Overall it’s a good work and contributes to the field of automation in aquaculture.

Reviewer 4 ·

Basic reporting

Overall, the manuscript aligns well with the standards set by PeerJ. Additionally, it offers a thorough and detailed overview of the relevant tools and algorithms employed, demonstrating a solid understanding of the current methodologies in the field. This comprehensive approach ensures that the manuscript meets the journal's expectations for rigor and clarity.

The Introduction and related work sections are well-presented and contextualized, effectively situating the study within the existing body of research. The manuscript is also well-referenced, providing a robust framework of citations that support its claims and methodology. The figures and tables included in the manuscript are highly relevant and contribute meaningfully to the overall narrative. They are well-labeled, of high quality, and clearly described within the text, enhancing the reader's understanding of the study's findings and conclusions.

The English language used throughout the manuscript is generally clear and understandable. However, to further enhance readability, the following revisions are suggested:
1. Introduce the abbreviation "IMC" for "intertidal mudflat culture" at its first mention and use the abbreviation consistently throughout the text.
2. Avoid repeating the phrase "in this study" within the same paragraph once it has already been mentioned at the beginning. For example, see lines 151-163.
3. Instead of referring to "the above equation," use a more specific reference like "Equation (1)" when discussing equations. This change should be applied to instances such as those found on lines 184 and 188

Experimental design

The manuscript seeks to introduce an enhancement to the YOLOv8 model, which is then applied to assist in razor clam fishing within intertidal mudflat culture (IMC). The proposed improvements to YOLOv8 include incorporating or adopting an IAIFI module, replacing the standard Bottleneck structure with the FasterNet Block, and implementing TBCAM. These modifications aim to optimize the model's performance in detecting and identifying razor clams in the IMC environment, thereby enhancing the efficiency and effectiveness of the fishing process.

The research questions are clearly defined and effectively address the existing gaps in the current body of knowledge. By focusing on these specific questions, the manuscript contributes valuable insights to the field and advances our understanding of the topic.

The field data that has been collected is sufficient to enable effective training of the model and to conduct a comprehensive evaluation of its performance. This dataset provides the necessary variety and volume of information required to accurately capture the underlying patterns and relationships, ensuring that the model can learn effectively and generalize well to new, unseen data. Additionally, the data allows for thorough testing and validation, enabling a robust assessment of the model's accuracy, reliability, and overall effectiveness in the applications.

The methods described were presented in a detailed and clear manner, ensuring that they can be easily followed and replicated by others. Each step of the process was carefully outlined, providing comprehensive instructions and explanations that reduce ambiguity and enhance understanding. This thorough documentation facilitates reproducibility, allowing to accurately replicate the study or experiment and verify the results.

Experiments were also conducted on publicly available datasets, and the results of these experiments are presented in the manuscript. The use of public datasets allows for broader validation of the findings and facilitates comparison with other studies in the field. Detailed descriptions of the experimental setup, including the specific datasets used and the methodologies applied, are provided to ensure transparency. The results are analyzed and discussed, highlighting the model's performance across various scenarios and demonstrating its applicability and effectiveness in diverse contexts.

Validity of the findings

The field data was utilized to run the improved IMC-YOLO model, and the results, presented as performance indicators, demonstrate that the model is robust, statistically sound, and accurate. The results show that the model effectively handles the complexities and variations inherent in data, maintaining high levels of precision and reliability.

The conclusions drawn from the study are well-aligned with the original research questions and are consistent with the findings related to the performance of the IMC-YOLO model. The study effectively addresses the objectives set forth at the outset by providing clear and comprehensive answers based on the model's results. The conclusions are clearly articulated, highlighting how the IMC-YOLO model meets the intended goals and contributes to advancing knowledge in the field. This alignment underscores the rigor and relevance of the research, demonstrating that the findings are directly connected to the questions posed.

Additional comments

The manuscript presents a study that addresses a significant real-world problem. It effectively utilizes an existing model (YOLO8v), implements improvements (IMC-YOLO), and rigorously tests the enhanced version. This commendable effort demonstrates the potential of the model to be applied to similar challenges in various practical scenarios. The approach and findings outlined in the manuscript provide valuable insights and practical solutions that could be replicated in analogous contexts.

---

## Round 0.2 · Minor Revisions

Dear authors,

Thank you for your paper. According to one reviewer, your paper still needs revision and we encourage you to address the concerns and criticisms of Reviewer 1 and resubmit your article once you have updated it accordingly.

Best wishes,

Reviewer 1 ·

Basic reporting

The authors should elaborate on the points raised. The innovative aspect of the paper is unfortunately not emphasised by the mathematical background. The authors are strongly encouraged to revise the paper extensively.

Experimental design

The authors should elaborate on the points raised. The innovative aspect of the paper is unfortunately not emphasised by the mathematical background. The authors are strongly encouraged to revise the paper extensively.

Validity of the findings

The authors should elaborate on the points raised. The innovative aspect of the paper is unfortunately not emphasised by the mathematical background. The authors are strongly encouraged to revise the paper extensively.

Reviewer 2 ·

Basic reporting

Well revised.

Experimental design

Well revised.

Validity of the findings

Well revised.

---

## Round 0.3 · Major Revisions

Dear author,

Thank you for the revision. Although your paper seems improved in some aspects, respected Reviewer 1 still has serious concerns and thinks that his/her criticisms have not been addressed. We encourage you to make the necessary additions and changes suggested by reviewer 1, particularly in terms of methodology, theoretical justification, comprehensive evaluation in terms of quality metrics, impact and clarity.

Best wishes,

Reviewer 1 ·

Basic reporting

The paper proposes minor modifications to the YOLOv8 architecture to detect razor clam burrows but fails to present a significant contribution to the field of object detection. The claimed improvements in performance metrics are minimal and do not justify the introduction of a new model. Furthermore, the paper lacks real-world validation, fails to provide strong theoretical justification for the modifications, and suffers from unclear writing. Given the incremental nature of the work and its failure to demonstrate real-world impact, this paper should be rejected. Significant improvements in methodology, evaluation, and clarity are necessary before reconsideration.

Experimental design

1) The proposed method, IMC-YOLO, appears to be an incremental modification of the existing YOLOv8 model. The modifications, such as the introduction of the Iterative Attention-based Intrascale Feature Interaction (IAIFI) module and FasterNet Block, do not represent a significant advancement in the field. These changes are superficial and do not introduce any groundbreaking innovation. The paper fails to present a compelling reason for why these modifications are necessary or how they fundamentally change the object detection task in the context of razor clam fishing.
2) The paper presents improvements in metrics like mAP50, mAP50:95, and F1best, but the improvements (2.2%, 3.5%, and 2.4%) are minimal and hardly justify the introduction of a new detection model. These marginal improvements are not enough to warrant the adoption of a new model over existing, well-established methods. Furthermore, the performance metrics alone do not adequately demonstrate the real-world applicability of the model, particularly when the improvements are this small.

Validity of the findings

3) The paper discusses improving fishing efficiency and reducing environmental damage, but there is no real-world validation to support these claims. The work focuses solely on the technical aspects of object detection without addressing the ecological or environmental impacts of implementing the proposed method. The authors must demonstrate how this method reduces environmental damage in practice. Without such validation, the claims made in the introduction remain unsubstantiated.
4) While the paper claims to compare IMC-YOLO with other mainstream object detection models, the comparison is superficial and lacks depth. There is no detailed analysis of how these models perform under similar conditions or why IMC-YOLO would be a preferable choice. A more thorough comparison, including a breakdown of computational cost, speed, scalability, and flexibility of use in various real-world conditions, is necessary.

---

## Round 0.4 · accepted · Accept

Dear Author,

Thank you for clearly addressing the reviewer's comments. Your paper seems now sufficiently improved after the last revision. Your manuscript is ready for publication.

Best wishes,

Reviewer 1 ·

Basic reporting

My comments have been addressed. It is acceptable in the present form.

Experimental design

My comments have been addressed. It is acceptable in the present form.

Validity of the findings

My comments have been addressed. It is acceptable in the present form.